# Machine Learning in Antibody Diagnostics for Inflammatory Bowel Disease Subtype Classification

**DOI:** 10.3390/diagnostics13152491

**Published:** 2023-07-26

**Authors:** Christiane Sokollik, Aurélie Pahud de Mortanges, Alexander B. Leichtle, Pascal Juillerat, Michael P. Horn

**Affiliations:** 1Division of Pediatric Gastroenterology, Hepatology and Nutrition, University Children’s Hospital, Inselspital, University of Bern, 3010 Bern, Switzerland; christiane.sokollik@insel.ch; 2ARTORG Center for Biomedical Engineering Research, University of Bern, 3010 Bern, Switzerland; aurelie.pahuddemortanges@unibe.ch; 3Department of Clinical Chemistry, Inselspital, Bern University Hospital, University of Bern, 3010 Bern, Switzerland; alexander.leichtle@insel.ch; 4Center for Artificial Intelligence in Medicine (CAIM), University of Bern, 3010 Bern, Switzerland; 5Department of Gastroenterology, Clinic for Visceral Surgery and Medicine, Inselspital, Bern University Hospital, University of Bern, 3010 Bern, Switzerland; pjuillerat@gesb.ch; 6Crohn’s and Colitis Center, Gastroenterology Beaulieu SA, 1004 Lausanne, Switzerland

**Keywords:** Crohn’s disease, ulcerative colitis, PR3-ANCA, serology, ASCA

## Abstract

Antibody testing in inflammatory bowel disease (IBD) can add to diagnostic accuracy of the main subtypes Crohn’s disease (CD) and ulcerative colitis (UC). Whether modern modeling techniques such as supervised and unsupervised machine learning are of value for finer distinction of subtypes such as IBD-unclassified (IBD-U) is not known. We determined the antibody profile of 100 adult IBD patients from the Swiss IBD cohort study with known subtype (50 CD, 50 UC) as well as of 76 IBD-U patients. We included ASCA IgG and IgA, p-ANCA, MPO- and PR3-ANCA, and xANCA measurements for computing different antibody panels as well as machine learning models. The AUC of an optimized antibody panel was 85% (95%CI, 78–92%) to distinguish CD from UC patients. The antibody profile of IBD-U patients was closely related to UC. No specific antibody profile was predictive for IBD-U nor for re-classification. The panel diagnostic was in favor of UC reclassification prediction with a correct assignment rate of 69.2–73.1% depending on the cut-off applied. Supervised machine learning could not distinguish between CD, UC, and IBD-U. More so, unsupervised machine learning suggested only two distinct clusters as a likely number of IBD subtypes. Antibodies in IBD are supportive in confirming clinical determined subtypes CD and UC but have limited capacity to predict IBD-U and reclassification during follow-up. In terms of antibody profiles, IBD-U is not a distinct subtype of IBD.

## 1. Introduction

The term inflammatory bowel disease (IBD) summarizes a spectrum of chronic diseases characterized by recurrent episodes of intestinal inflammation. There are two main subtypes: Crohn’s disease (CD) and ulcerative colitis (UC). In recent years, it was proposed to better describe the continuum within IBD by dividing the classification more precisely [1,2]. Herewith, special attention is paid to CD patients with isolated colonic disease location [3,4,5] as well as patients where no classification is possible [6,7]. For the latter patient group, the term IBD-unclassified (IBD-U) is established. In different cohorts, 6–13% of IBD patients are labeled as IBD-U [8,9]. Clinically, there is evidence that a finer distinction of IBD subtypes may be important for prognosis and management strategies [5,9]. However, it is still not clear whether IBD-U is a distinct disease entity of IBD or a milder and earlier stage of CD or UC. Cleynen et al. developed a CD versus UC genetic risk score, which placed colonic CD as well as colonic IBD-U between ileal CD and UC [1]. On the other hand, antibody testing placed colonic CD closer to CD and IBD-U closer to UC [5,7].

Previously, we have shown that antibody-based panel diagnostics was superior to single antibody testing in distinguishing between CD and UC in pediatric IBD patients [10]. In this study, we performed antibody diagnostics in a cohort of adult IBD patients with known CD and UC as well as on IBD-U patients. We validated the classification power of antibodies employing modern modeling including supervised and unsupervised machine learning. Furthermore, we tested different modeling approaches in IBD-U patients without and with reclassification during follow-up.

## 2. Materials and Methods

### 2.1. Study Population and Design

We enrolled 176 IBD patients (50 CD, 50 UC, and 76 IBD-U patients) of the Swiss IBD Cohort Study (SIBDCS). The SIBDCS prospectively follows IBD patients with yearly-standardized follow-ups, which combine clinical data collection and bio-sampling [11,12]. Collected clinical data include sex as well as age at diagnosis, enrollment, follow-up, and serum sampling. The diagnosis of IBD was based according to international standards on a combination of clinical, biochemical, stool, endoscopic, and histological examinations [13]. The Montreal classification was used for clinical phenotyping including IBD-U, and the UC nomenclature was used for disease location of IBD-U patients [14,15]. IBD-U patients had no definitive histological or other evidence, which was in favor of either CD or UC. Patients with isolated colonic disease were subordinated to the CD group. A subset of 20 IBD patients with a definite diagnosis at the last follow-up was reviewed by an independent gastroenterologist (F.B.) for the appropriateness of the classification or reclassification during follow-up using available endoscopic, histological, and radiological reports. He was blinded to the diagnoses documented in the SIBDCS database.

All sera included in the analyses were sampled after inclusion into the study. In the subgroup of IBD-U patients who were re-classified during follow up, serum sampling always took place before reclassification. At time of serum sampling, all patients had already received treatment at the discretion of their physician.

The Swiss IBD cohort study has been approved by the local ethics committee of each participating center (institutional review board No. EK-1316, approved on 5 February 2007 and KEK of Canton of Zurich, 2018–02068 on 9 March 2020). Patients gave written informed consent for inclusion in the SIBDCS. The study was conducted in accordance with the Helsinki declaration. 

### 2.2. Analysis of Antibodies

All sera were analyzed in a blinded fashion without knowledge of patient diagnosis or other clinical information. We determined the antibodies as previously described [10]. All tests were performed according to the manufacturer’s instructions and were carried out in our diagnostic routine laboratory. Commercial tests and normal reference values as used in routine diagnostic were applied: ASCA IgA and IgG < 7 U/mL (EliA, Thermo Fisher Diagnostics (Reinach, Switzerland)); PR3-ANCA < 5.0 U/mL, MPO-ANCA < 6.0 U/mL (both CLIA QuantaFlash, INOVA Dx (San Diego, CA, USA)); indirect immunofluorescence titers of xANCA, cANCA, and pANCA (EtOH fixed neutrophil granulocytes, INOVA Dx); as well as ANA < 1:80 (HEp-2; INOVA Dx). All atypical pANCA reactions (defined as pANCA reaction on EtOH fixed neutrophil granulocytes without MPO-ANCA) were set to negative, if the ANA titer was higher than the ANCA titer, as the pANCA reaction was disturbed by ANA. 

### 2.3. Statistical Analysis

All analyses were performed using SAS software, version 9.3 (SAS Institute, Cary, NC, USA) or R, version 4.2.2 (The R Foundation, Vienna, Austria). A *p*-value < 0.05 was considered statistically significant for all analyses.

Univariate analysis was performed using Wilcoxon rank score test for continuous variables and Chi square or Fisher’s exact test for categorical variables. Sensitivity and specificity were calculated for the antibody panels in order to predict the disease subtype. 

A quasi-exhaustive logistic regression approach on a predefined set of antibody data was applied to find the best discriminative model for CD and UC. Based on the Akaike Information Criterion (AIC, Appendix A), an efficient branch-and-bound algorithm was used for the exhaustive search for the best subsets of the variables [16]. The dichotomous diagnosis class (CD vs. UC) was used as dependent and the antibodies as independent variables (R v.4.2.2), and the model space was assessed using ‘leaps’-based (package ‘leaps_3.0′) wrapper functions. ROC curves were drawn with the pROC package (v. 1.9.1), and the computation of optimal cutoffs was done with Youden’s J statistic (package pROC). 

Using Bayesian Model Averaging (BMA) for binomial logistic regression, we extracted the variables with the highest predictive importance for the distinction between UC and CD according to their inclusion probability into the generated models. In this analysis, patients who were reclassified from IBD-U to UC or CD by the end of the follow-up period were also included. Bayesian Model Averaging was conducted with the BMA package.

After investigating prediction of IBD patients for two classes, we also conducted a multinomial logistic regression with a three-class outcome, UC, CD, and IBD-U. Additionally, an XGBoost algorithm was trained for the three-class disease prediction to explore potential added value of more advanced machine learning techniques over logistic regression. For both three-class prediction models, the data were randomly split into training and test subsets in a 66:34 ratio, to ensure adequate representation of all three classes in both the training and the test subsets derived from our relatively small overall data set. Both models were trained with 5-fold internal cross-validation and created with the caret package.

To scrutinize whether a three-class model of IBD (CD vs. UC vs. IBD-U) is supported by the underlying data, we conducted k-means clustering, an unsupervised learning technique, in which n observations are partitioned into a pre-specified number of k clusters according to the nearest cluster mean. In concordance to the three-class hypothesis, we conducted k-means clustering with k = 3. We also sought to determine the number of clusters suggested by the data using the “elbow method” (based on within-cluster-sum of squared errors) as well as by the “silhouette method” (based on the silhouette coefficient). Additionally, general agglomerative hierarchical clustering was done applying Ward’s minimum variance method. Here the objective function, depending on which clusters are merged, is the error sum of squares. A corresponding heatmap of the clustering was created. 

K-means clustering and ward hierarchical clustering were performed using the stats package, and pheatmap was used for heatmap creation. 

## 3. Results

### 3.1. Antibody Status and Panel Diagnostic in Adult CD and UC Patients

An overview of clinical data and single antibody results of CD and UC patients are provided in Appendix A. 

ASCA IgA and IgG antibodies were highly specific for CD (*p* < 0.001 and *p* < 0.001, respectively) whereas xANCA and PR3-ANCA were specific for UC (*p* < 0.001 and *p* = 0.006, respectively). The positivity of cANCA and (atypical) pANCA did not discriminate between the CD and UC. None of the analyzed serum samples was positive for MPO-ANCA.

We first applied our previously described antibody panel [10] (named panel BeLu) for its classification power in adult IBD patients. The panel BeLu correctly assigned 37/50 CD patients (74%) and 32/50 UC patients (64%), respectively. This adds up to 69/100 (69%) correct assignments with an AUC of 78% (95%CI, 69–87%) (Figure 1, red line).

Next, we used all determined antibodies as the input for computing an optimized panel for adult patients (named panel SIBDCS). The resulting model selected only the positivity status of PR3-ANCA, xANCA, ASCA IgA, and ASCA IgG. Intercept and the factors of the single antibody results are shown in Table 1. Using this panel, the correct assignment rate increased in comparison to the panel BeLu to 77/100 (77%) with an AUC of 85% (95%CI, 78–92%) (Figure 1, blue line). Three different cut-offs, 0.61, 0.62, and 0.65, were calculated to be optimal for the distinction between CD and UC depending on the priority which subtype should be predicted. By adapting the predictor cut-off from 0.61 to 0.62 and 0.65, the specificity for the classification of CD increased from 88% (95%CI, 78–96%) to 92% (95%CI, 84–98%) and 94% (95%CI, 88–100%), respectively. However, this adaptation lowered the sensitivity for CD from 66% (95%CI, 52–78%) to 62% (95%CI, 48–76%) and 60% (95%CI, 46–74%), respectively. 

### 3.2. Antibody Status and Panel Diagnostics in IBD-U Patients 

Next, we evaluated the predictive capacity of antibodies and derived panels for the classification of IBD-U patients. We determined the same antibodies as above in 76 IBD-U patients of the SIBDCS (Table 2 and Table 3). Furthermore, 46/76 (60.5%) of IBD-U patients were negative for PR3-ANCA as well as ASCA. In general, the antibody status of IBD-U patients was closely related to UC with a high number of xANCA (46/76 (60.5%) vs. 25/50 (50%) UC patients, respectively) and PR3-ANCA (20/76 (26.3%) vs. 12/50 (24%), respectively) positivity as well as a low number and low titer of ASCA positivity (14/76 (18.4) vs. 34/50 (68%) CD patients). Independently of the applied panel—BeLu or SIBDSC—the panel diagnostic favored UC classification in most IBD-U patients as well (Figure 2a,b).

During follow-up, 26/76 (34.2%) of the IBD-U patients were reclassified. Of these 26 patients, 8 reclassified to CD (30.8%) and 18 to UC (69.2%). Furthermore, 50 IBD-U patients were not reclassified during follow-up time of mean 6.5 y (IQR 3.9–12.1 y).

The reclassification to CD was associated with ASCA-IgA positivity in 2/8 patients (25.0%), and the reclassification to UC was associated with xANCA positivity in 11/18 (61.1%) and PR3-ANCA positivity in 7/18 (38.9%) patients. 

The panel BeLu correctly assigned 19/26 (73.1%) of the reclassified IBD-U patients: 5/8 (62.5%) of those who reclassified to CD and 14/18 (77.8%) of those who reclassified to UC (Figure 2a). The panel SIBDCS correctly predicted the reclassified subtype in 18/26 (69.2%) IBD-U patients when using the cut-off of 0.65 (sensitivity 60% and specificity 94%) (Figure 2b). The panel assigned correctly 1/8 (12.5%) who reclassified to CD and 17/18 (94.4%) who reclassified to UC. When applying the lower cut-offs of 0.62 or 0.61, the accuracy for CD remained at 1/8 patients, but the correct prediction rate for UC dropped to 15/18 patients. For reclassification of the IBD-U patients, both panels performed with similar accuracy.

### 3.3. Supervised and Unsupervised Machine Learning for Bi- and Multiclass Prediction Models (CD vs. UC and CD vs. UC vs. IBD-U)

After developing the logistic regression models, BeLu and SIBDCS, we investigated in different exploratory analyses whether modern machine learning techniques offer added value. As an extension of the above-presented logistic regression-based models, we conducted BMA, as model averaging procedures provide better predictive performance in the presence of model uncertainty [17]. In the analysis, more than 60 models were generated, showing an overall accuracy of 74.6% (sensitivity: 88.2%, specificity: 58.6%) for the distinction between UC and CD when calculated from the whole dataset without holdouts. The parameters “xANCA (0/1)”, “PR3-ANCA (0/1)”, and “any ASCA (0/1)” were most frequently included in models and could therefore be assumed to hold the most predictive power (Figure 3).

After exploring various models with the binary outcome CD vs. UC, we were also interested in multinomial approaches classifying CD vs. UC vs. IBD-U, following the hypothesis that IBD-U might be its own class, distinct from CD and UC. Both three-class multinomial logistic regression as well as XGBoost prediction algorithm did not offer added classification performance advantages compared to the two class exhaustive logistic regression models presented above. Multinomial logistic regression yielded an overall accuracy of 65% for the training subset with sensitivities for the three classes ranging from 39–77% and specificities of 72–88%. On the testing subset, the performance dropped to 48% (sensitivities: 39–63%, specificities: 63–79%). Similarly, XGBoost algorithm yielded an out-of-fold prediction accuracy of 76% (sensitivities: 58–87%, specificities: 82–92%) and a test accuracy of 70% (sensitivities: 52–63%, specificities: 74–81%).

The three-cluster k-means clustering showed a low overall accuracy of 41%, exhibiting specificities between 57% and 90% and sensitivity from 0% to 64% for the three classes CD, UC, and IBD-U. The sensitivity for IBD-U was 0%. This difficulty of separating IBD-U cases is demonstrated in Figure 4, where the visualization of the clustering showed that CD cases separate rather distinctly, while UC and IBD-U cases exhibit a large overlap. Visualization of the optimal number of clusters through the “elbow method” (based on Within-Cluster-Sum of Squared Errors) as well as through the “silhouette method” (based on the silhouette coefficient) indicates that the true number of clusters is more likely to be two than three (Appendix A). Ward hierarchical clustering and the corresponding heatmap also indicated poor clustering based on a three-class diagnosis-based model (Figure 5A,B).

## 4. Discussion

Our study was set up to test the predictive value of antibody diagnostics for classification in IBD. Employing our previously established panel BeLu [10], we reached an AUC 77% in an adult cohort for correct classification of CD and UC. We could even improve the AUC to 85% with an optimized panel SIBDCS, which only used the positivity or negativity of four widely available antibodies (PR3-ANCA, xANCA, ASCA IgA and ASCA IgG). 

We did not find a specific antibody profile to reliably classify IBD-U patients. Our results of supervised machine learning point in the direction that antibody testing is valuable for binomial distinction in CD and UC but lacks capability for finer tuning. Furthermore, unsupervised machine learning suggested that the true underlying number of clusters generatable by antibody testing is more likely to be two than three. Following this, it can be deduced that, indeed, IBD-U is from an antibody point of view not a distinct disease entity of IBD. 

In our cohort, 26/76 (34.2%) IBD-U patients were reclassified during follow-up: 8/26 (30.8%) to CD and 18/26 (69.2%) to UC. These rates are in the same range as described in previous cohorts studied with reclassification rates between 21 and 54%. Of the reclassified patients, 28–83% were reclassified to CD and 17–71% to UC [2,7,9,18]. The wide range of the reclassification rates may be explained by selection bias of the cohorts including age of diagnosis or length of follow-up. For example, Rinawi et al. [18] reported a median follow-up at reclassification to CD of 9.4 years, whereas most other studies had a much shorter median follow-up time. The unbalanced distribution between CD and UC reclassification with a preferable reclassification to UC has to be seen as a reflection of natural history but also may argue in favor of IBD-U as an early stage of IBD [7,9]. 

There are two previous studies investigating the predictive value of antibody profiles for reclassification [19,20]. In their studies, Joossens et al. and Birimberg-Schwartz et al. assessed ASCA, pANCA (whereas pANCA also included atypical pANCA and xANCA), and their combinations. They found that pANCA negativity in combination with ASCA negativity was most prevalent in IBD-U and associated with the least probability of reclassification during follow-up. In our study, we found that PR3-ANCA negativity in combination with ASCA negativity was the most prevalent antibody combination in IBD-U patients independent of whether they later were reclassified or not. These findings may therefore reflect a true IBD-U characteristic. However, they could also be conditioned by biased physicians, who unconsciously add antibody results in their classification algorithms. 

Fitting with our previous observation that PR3-ANCA positivity is predictive of UC in pediatric IBD patients [10], PR3-ANCA positivity was associated with reclassification to UC. In addition, our Bayesian model averaging identified PR3-ANCA to be used in most models for classification of CD vs. UC. In the same line, we could show for panel diagnostics a low sensitivity for reclassification to CD and a better performance for correct prediction of reclassification to UC in IBD-U patients. 

Reflecting all results, antibody testing supports the distinction into two subtypes of IBD. One argument to focus on finer distinction of subtypes was, for example, the introduction of biological treatments with a benefit of early anti-TNF alpha treatment in CD patients [21]. However, with newer biologicals, which are efficiently independent of the IBD subtype, this argument may again fade into the background. Indeed, clinical factors (e.g., age or frailty of the patient, comorbidities, genetic background) as well as predictive molecular markers specific to the therapeutic agent will become more important for the choice of treatment [22].

We acknowledge the small sample size as the main limitations of our study. This prohibited deeper analysis of isolated colonic CD patients, which we included in the CD group. A strength is the inclusion of all IBD-U patients from the SIBDCS, which has a good representation of IBD in Switzerland. The use of well-established antibody markers, which can be determined in most immunology routine laboratories, ameliorates the utility of antibody diagnostics. However, a general challenge in antibody testing is the analytical method. For example, Mahler et al. showed for PR3-ANCA a positive rate of 88/283 UC samples with chemiluminescence and of 17/283 with ELISA [23]. 

## 5. Conclusions

Our study shows that antibody diagnostics are helpful for CD vs. UC classification but have limited potential in finer subtyping of IBD or predicting reclassification of IBD-U patients. New modeling techniques based on antibodies support the classification in two main IBD subtypes.

## Figures and Tables

**Figure 1 diagnostics-13-02491-f001:**
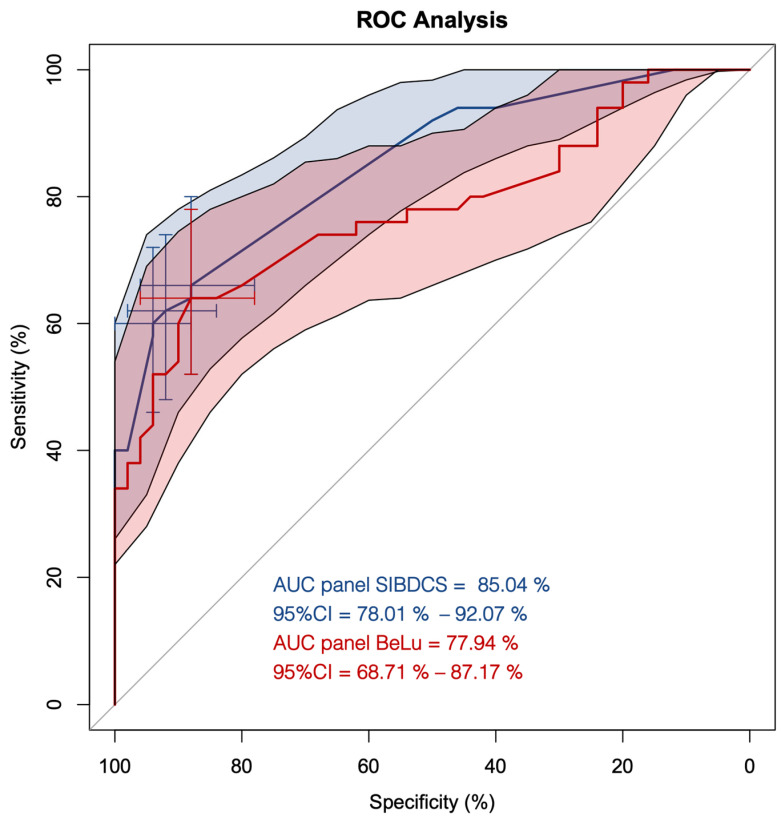
ROC curve analysis of the antibody panels BeLu and SIBDCS in patients diagnosed with Crohn’s disease (CD) and ulcerative colitis (UC). Red line with red shaded area, ROC curve of panel BeLu +/− 95%CI; blue line with blue shaded area, ROC curve of panel SIBDCS +/− 95% CI; cross hairs, optimized cut-offs distinguishing CD from UC.

**Figure 2 diagnostics-13-02491-f002:**
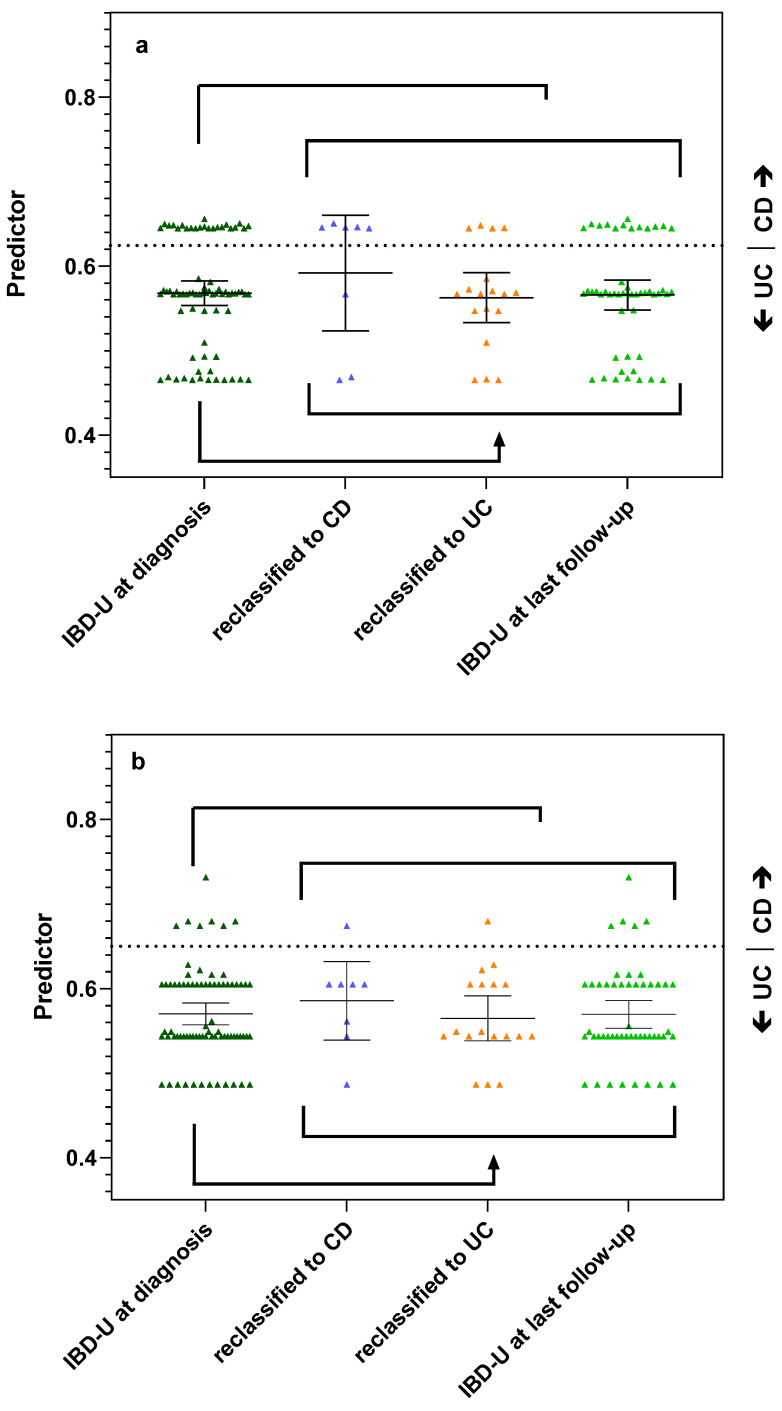
Antibody panels predicting reclassification of IBD-U patients during follow-up. (**a**) Performance of the BeLu panel. Dashed line indicates previously published cut-off 0.63 [10]. (**b**) Performance of the SIBDCS panel. Dashed line indicates optimized cut-off of 0.65 with a sensitivity of 60% and a specificity of 94% for the classification of CD. Each symbol represents the calculated predictor value of a patient. Lines and whiskers indicate mean and 95% CI of the different patient groups.

**Figure 3 diagnostics-13-02491-f003:**
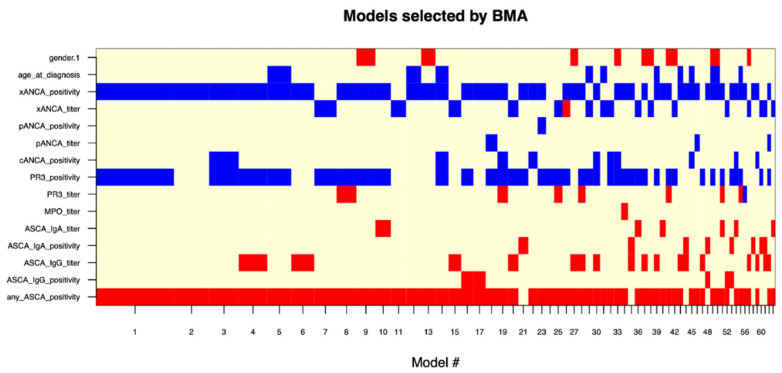
Overview of Bayesian model averaging models. Over 60 were generated and ordered by decreasing importance from left to right on the x-axis. Red and blue bars indicate inclusion and sign of the variable into a certain model. Variables with large continuous bars can be read to be included in many models and are therefore very likely to hold high predictive value. BMA: Bayesian model averaging.

**Figure 4 diagnostics-13-02491-f004:**
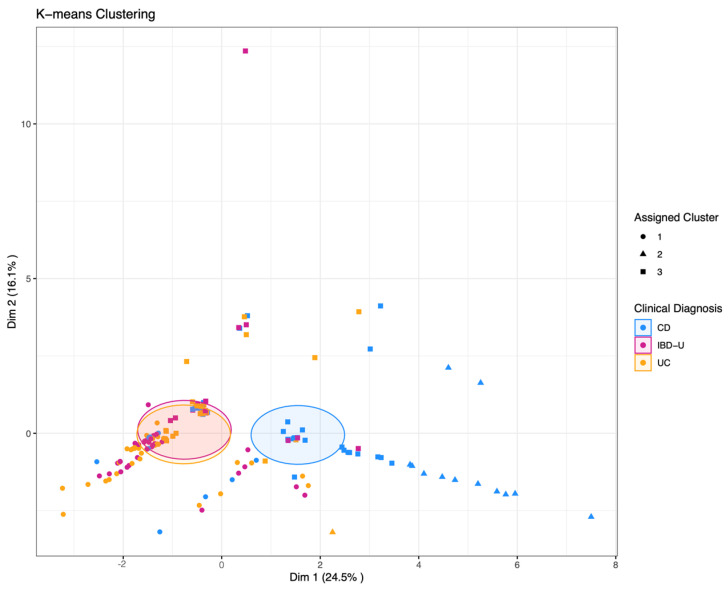
K-means clustering with k = 3 pre-specified groups. The data points each represent one patient. Point shapes indicate the assigned cluster one to three; the colors indicate the clinical diagnosis of CD, UC, or IBD-U, respectively. Shaded areas indicate the Euclidean distance from the group center.

**Figure 5 diagnostics-13-02491-f005:**
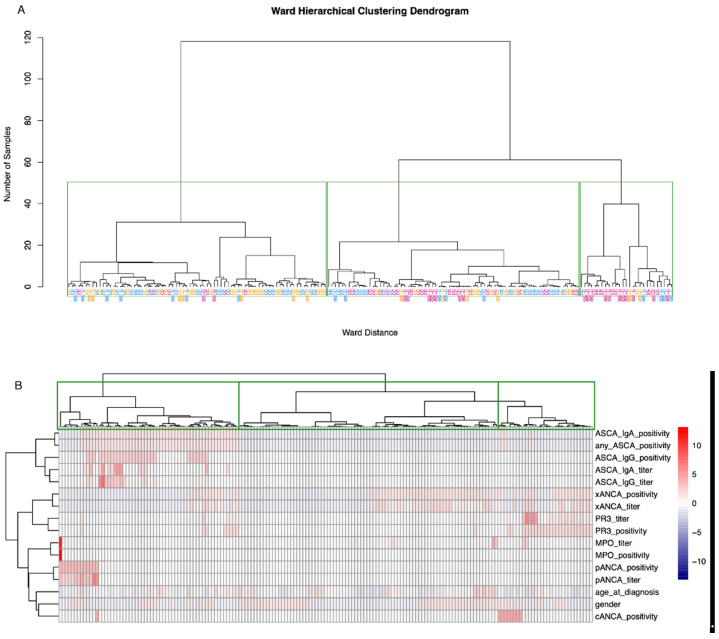
Ward hierarchical clustering and corresponding heatmap. (**A**) Green rectangles are drawn around clusters in the dendrogram as found by the ward hierarchical clustering algorithm. CD, UC, and IBD-U labels indicate the true disease class membership. The colors of the class labels correspond to the classes as found by the k-means clustering. Blue = cluster 1, magenta = cluster 2, orange = cluster 3. (**B**) The dendrogram on the *x*-axis, including the rectangles around clusters, above the heatmap corresponds to figure (**A**). The dendrogram on the *y*-axis represents the grouping of predictor variables. Values are scaled per parameter.

**Table 1 diagnostics-13-02491-t001:** Panel description.

Analyte	BeLu *	SIBDCS
Intercept	0.5937	0.425704
PR-3 ANCA positivity	−0.4085	−0.22856
xANCA positivity	−0.328	−0.25171
pANCA positivity	−0.6299	n.a.
ASCA IgG, Titer	0.0052	n.a.
ASCA IgG positivity	n.a.	0.277831
ASCA IgA positivity	n.a.	0.301052

* BeLu (Bern Lucerne) panel taken from [10]; n.a. not applied; SIBDCS Swiss IBD cohort study.

**Table 2 diagnostics-13-02491-t002:** Characteristics of IBD-U patients.

	All IBD-U		IBD-U w/o Reclassification		IBD-U w/Reclassification	
number of patients	76		50		26	
males, *n* (%)	38 (50)		24 (48)		14 (53.8)	
age at diagnostic, median (IQR), y	20 (12–31)		23 (13–37)		17.5 (11–28)	
age at serum sampling, median (IQR), y	23.5 (14–40)		28 (14–46)		17.5 (12–30)	
disease duration at serum sampling, median (IQR), y	2 (1–5)		3 (1–6.5)		1 (0–5)	
disease duration at last follow-up, median (IQR), y	6.5 (4–12)		7 (4–11)		6 (4–16)	
disease duration at reclassification, median (IQR), y	n.a.		n.a.		3.5 (3–9)	
need of surgery, *n* (%)	6 (7.9)		3 (6.0)		3 (11.5)	
ever treated with biologicals, *n* (%)	43 (56.6)		27 (54)		16 (61.5)	
**Disease location at diagnosis and last follow-up, n (%)**	**diagnosis**	**follow-up**	**diagnosis**	**follow-up**	**diagnosis**	**follow-up**
E1: proctitis	3 (3.9)	4 (5.3)	2 (2.6)	1 (1.3)	1 (1.3)	3 (3.9)
E2: left-sided colitis	15 (19.7)	18 (23.7)	11 (14.5)	13 (17.1)	4 (5.3)	5 (6.6)
E3: extensive (pancolitis)	41 (53.9)	41 (53.9)	24 (31.6)	30 (39.5)	17 (22.4)	11 (14.5)
unknown	17 (22.4)	6 (7.9)	13 (17.1)	6 (7.9)	4 (5.3)	0 (0)
L1: ileal	n.a.	1 (1.3)	n.a.	n.a.	n.a.	1 (1.3)
L2: colonic	n.a.	4 (5.3)	n.a.	n.a.	n.a.	4 (5.3)
L3: ileo-colonic	n.a.	0 (0)	n.a.	n.a.	n.a.	0 (0)
L4: upper GI disease	n.a.	0 (0)	n.a.	n.a.	n.a.	0 (0)
no endoscopy	n.a.	2 (2.6)	0 (0)	0 (0)	0 (0)	2 (2.6)

IQR: interquartile range. y: years. n.a.: not applicable.

**Table 3 diagnostics-13-02491-t003:** Antibody status of IBD-U patients.

	All IBD-U	IBD-U w/o Reclassification	IBD-U w/Re- Classification *	IBD-U → CD *	IBD-U → UC *
number of patients	76	50	26	8	18
**ANCA positive**					
cANCA, *n* (%)	5 (6.6)	4 (8.0)	1 (3.8)	0 (0)	1 (5.6)
(atypical) pANCA, *n* (%)	3 (3.9)	3 (6.0)	0 (0)	0 (0)	0 (0)
xANCA, *n* (%)	46 (60.5)	32 (64.0)	14 (53.8)	3 (37.5)	11 (60.5)
PR3-ANCA, *n* (%)	20 (26.3)	11 (22.0)	9 (34.6)	2 (25.0)	7 (38.9)
PR3-ANCA U/mL, Median, IQR	1.3 (0–5.2)	0.9 (0.4–3.1)	1.9 (0.8–6.9)	1.2 (0.5–6.2)	2.8 (1.0–7.0)
MPO-ANCA, *n* (%)	1 (1.3)	1 (2.0)	0 (0)	0 (0)	0 (0)
MPO-ANCA U/mL, Median	0	0	0	0	0
**ASCA positive**					
IgA, *n* (%)	13 (17.1)	8 (16.0)	5 (19.2)	2 (25.0)	3 (16.7)
IgA U/mL, Median, IQR	2.0 (1–3.9)	2.3 (1–4)	1.7 (0–3.9)	2.5 (1.4–6.7)	1.6 (0–2.5)
IgG, *n* (%)	6 (7.9)	4 (7.9)	2 (7.7)	0 (0)	2 (11.1)
IgG U/mL, Median, IQR	1.4 (0.6–3.1)	1.5 (0.6–3.0)	1.1 (0.6–3.4)	1.3 (0.6–2.9)	0.8 (0.6–3.5)
**Antibody combinations (*n* (%))**					
xANCA neg, PR3-ANCA neg, all ASCA neg	21 (27.6)	13 (26.0)	8 (30.8)	4 (50.0)	4 (22.2)
xANCA pos, PR3-ANCA pos	15 (19.7)	9 (18.0)	6 (23.1)	2 (25.0)	4 (22.2)
xANCA neg, all ASCA neg	25 (32.9)	15 (30.0)	10 (38.5)	4 (50.0)	6 (33.3)
xANCA neg, any ASCA pos	5 (6.6)	3 (6.0)	2 (7.7)	1 (12.5)	1 (5.6)
xANCA pos, all ASCA neg	37 (48.7)	26 (52.0)	11 (42.3)	2 (25.0)	9 (50.0)
xANCA pos, any ASCA pos	9 (11.8)	6 (12.0)	3 (11.5)	1 (12.5)	2 (11.1)
PR3-ANCA neg, all ASCA neg	46 (60.5)	31 (62.0)	15 (57.7)	5 (62.5)	10 (55.6)
PR3-ANCA neg, any ASCA pos	10 (13.2)	8 (16.0)	2 (7.7)	1 (12.5)	1 (5.6)
PR3-ANCA pos, all ASCA neg	16 (21.1)	10 (20.0)	6 (23.1)	1 (12.5)	5 (27.8)
PR3-ANCA pos, any ASCA pos	4 (5.3)	1 (2.0)	3 (11.5)	1 (12.5)	2 (11.1)

***** neither any of the tested antibody nor any antibody combination significantly distinguished reclassified patients from IBD-U patients w/o reclassification.

## Data Availability

The data underlying this article can be shared on reasonable request to the corresponding author.

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
