# Peer review of "Machine Learning in Antibody Diagnostics for Inflammatory Bowel Disease Subtype Classification"

_diagnostics, 2023, doi:10.3390/diagnostics13152491_

Round 1

Reviewer 1 Report

The authors present a new antibody-based biomarker panel SIBDCS for distinguishing CD and UC based on a 100 patient-cohort study and provide some insights to IBD-unclassified (IBD-U) patients using the new panel. First, I find the IBD-U class is vaguely defined. Is this class consisting unclassified IBD-U patients as well as colonic CD patients? For the SIBDCS panel predicted by the quasi-exhaustive logistic regression, a graphic plot of the model selection would be helpful too. Compared to the previous BeLu panel, the SIBDCS panel is not performing better when reclassifying the IBD-U samples. The performance of the Bayesian model averaging is unknown and is the purpose of this analysis only to gain the re-recognition of xANCA/PR3-ANCA/ASCA? This section seems disconnected in the manuscript. When the authors build multi-class predictor using XGBoost, the results are not so well. The K-means clustering is hard to see and would be better to color by the diagnosis. The authors claim that the optimal k cluster is 2, however, the referenced elbow and silhouette width plots do not support this claim. I think the authors need to address this better by considering other parameters, such as age and treatment.

Author Response

Replies to Reviewer 1

Comments and Suggestions for Authors

All lines indicated rely to the revised version in the change track mode.

The authors present a new antibody-based biomarker panel SIBDCS for distinguishing CD and UC based on a 100 patient-cohort study and provide some insights to IBD-unclassified (IBD-U) patients using the new panel.

  1. First, I find the IBD-U class is vaguely defined. Is this class consisting unclassified IBD-U patients as well as colonic CD patients?

We thank the reviewer for this question.

IBD-U class was given to patients with features that make the differentiation between UC and CD uncertain. There are no standardized diagnostic criteria, but attempts to standardize classification (Birimberg-Schwartz J Crohns Colitis. 2017 Sep 1;11(9):1078-1084). However, adaptations and validations are still ongoing (Ledder JCC, 2020, 1672–1679). We therefore relied on the treating physicians’ classification and the review by an independent gastroenterologist (F.B.).

To make this point more clearly, we adapted the introduction on line 46:

“Herewith special focus is paid to CD patients with isolated colonic disease location [3-5] as well as patients where no classification is possible [6,7]. For the latter patient group the term IBD-unclassified (IBD-U) is established.”

Furthermore, we modified in the methods section on lines 70ff the phrase:

The Montreal classification was used for clinical phenotyping including IBD-U and the UC nomenclature was used for disease location of IBD-U patients [14,15]. IBD-U patients had no definitive histological or other evidence, which was in favor of either CD or UC. Patients with isolated colonic disease were subordinated to the CD group.

  1. For the SIBDCS panel predicted by the quasi-exhaustive logistic regression, a graphic plot of the model selection would be helpful too.

We thank the reviewer for this valuable suggestion. We plotted the model selection according to the Akaike Information Criterion and added the requested Figure to the supplementary material as Supplementary Figure 2.

Supplementary Figure 2: Akaike Information Criterion (AIC) profile plot. The horizontal line delineates models that are less than 2 AIC units away from the best model (on the very left). In total, 376 model combinations were assessed.

In the methods section on line 109 we added the link to supplementary figure 2.

  1. Compared to the previous BeLu panel, the SIBDCS panel is not performing better when reclassifying the IBD-U samples.

We thank the reviewer for this comment. We added in paragraph 3.2 on line 225 the phrase "For reclassification of the IBD-U patients, both panels performed with similar accuracy."

  1. The performance of the Bayesian model averaging is unknown and is the purpose of this analysis only to gain the re-recognition of xANCA/PR3-ANCA/ASCA? This section seems disconnected in the manuscript.

We thank the reviewer to raise this point. In order to provide more comprehensive information regarding the BMA analysis, we added its performance metrics.

The purpose of the BMA analysis was, together with the other machine learning methods applied, to gain exploratory insight into the utility of such algorithms when used for the prediction of IBD subtype from antibody panel results. We start the section on machine learning models with the BMA analysis, as it is conceptually most closely related to the quasi-exhaustive logistic regression previously described, as both analyses build individual units of logistic regression.

The re-recognition of xANCA/PR3-ANCA/ASCA was the result rather than the purpose of the analysis.

We have adapted the section in Paragraph 3.3 on line 230ff

After developing the logistic regression models, BeLu and SIBDCS, we investigated in different exploratory analyses whether modern machine learning techniques offer added value.

As an extension of the above-presented logistic regression-based models, we conducted BMA, as model averaging procedures provide better predictive performance in the presence of model uncertainty [new reference 17]. In the analysis, more than 60 models were generated, showing an overall accuracy of 74.6% (sensitivity: 88.2%, specificity: 58.6%) for the distinction between UC and CD when calculated from the whole dataset without holdouts. The parameters "xANCA (0/1)", "PR3-ANCA (0/1)", and "any ASCA (0/1)" were most frequently included in models and could therefore be assumed to hold the most predictive power. (Figure 3).

  1. When the authors build multi-class predictor using XGBoost, the results are not so well.

We agree with the reviewer that the performance metrics of the XGBoost model described in the first draft of the manuscript indeed indicate poor performance. The reviewers comment led us to reinvestigate and retune the XGBoost model with the caret package. We were able to improve training and testing results, mainly by an increase in sensitivity and a potential reduction of overfitting. The now resulting accuracy of 76.1% (CI 67.3% - 83.5%) for training and 56.9% (43.2% - 69.8%) for testing is plausible, as it is within the range of the other models. These new metrics have been adapted in the manuscript on line 258ff.

Additionally, it can be noted that in a multi-class classification problem with n classes, the accuracy of a random uninformative model is 1/n. Therefore, the “coin toss” probability of correct classification for a three-class model is 0.33, not 0.5. Higher accuracies are therefore harder to achieve.

  1. The K-means clustering is hard to see and would be better to color by the diagnosis.

We thank the reviewer for this suggestion to investigate a switching of colors and shapes in Figure 4 illustrating the k-means clustering. We tried out how the figure would look with this switch. (Please see below.) In our opinion, representing the clinical diagnoses with colors and shaded areas instead of the assigned clusters, changes the message of the Figure, shifting the focus away from the cluster to the diagnoses. Therefore, we decided to keep the initial assignment of shapes and colors.

To address the readability, we updated the Figure in the manuscript to a reduced point size of the symbols representing the clinical diagnoses. This may help to distinguish data points more easily.

Figure 4 with colors and shapes switched:

Figure 4 with reduced point size for shapes:

  1. The authors claim that the optimal k cluster is 2, however, the referenced elbow and silhouette width plots do not support this claim. I think the authors need to address this better by considering other parameters, such as age and treatment.

We thank the reviewer for pointing this out. We agree that the elbow and silhouette plots do not indicate 2 as the most likely number of clusters. In the manuscript, we intended to write that the true number of clusters is more likely to be 2 than 3. This is in concordance with the plots. During the editing of the manuscript, parts of this statement got lost thereby altering its meaning. We have now corrected the statement in the manuscript.

According to the plots, the “true” number of classes might be 9. This is not concordant with the current clinical understanding of IBD subtype classification, and we therefore do not discuss it further in our manuscript. The main message of the analysis was to demonstrate the higher likelihood of 2 classes compared to 3.

Regarding the inclusion of age and treatment: “Age at diagnosis” is included as a predictor in all our analyses including the clustering analyses. We do not include information about treatment in any of our analyses of this manuscript, as we intend to base the classification solely on objective parameters, such as serology testing, that are available ideally before treatment starts.

===================================

Reviewer 2 Report

 The author said that antibody diagnostics is helpful for CD vs UC classification, and this research showed  new modeling techniques based on antibodies support the classification in two main IBD subtypes. This paper covered many experiments between CD, UC or IBD-U patients. This paper was excellent and will be useful of research in CD, UC or IBD-U patients

I have no claim about the Quality of English language 

Author Response

We thank the Reviewer for the short feedback on the Manuscript. In order to improve the English language of the manuscript we will profit from the MDPI Editing Service if necessary.

===================================

Reviewer 3 Report

The authors previously developed and tested the BeLu antibody panels for children and adolescents with inflammatory bowel disease. In this work, the developed BeLu panel was used to classify adult patients. Moreover, the authors proposed a new panel SIBDCS with better sensitivity and specificity.

The study is well-designed and carried out as well. The authors applied for antibody measurement commercial tests and normal reference values as used in routine diagnostic. The only drawback of the work - and the authors write about it - the developed predictive model will have the characteristics described in this work only in the case of measuring autoantibodies using the specific tests used.

Minor:

Line 57 “during follow-up..” - Please check extra punctuation marks

In Table 1 words are far apart: . “IBD-U” and “w/o reclassification”, “IBD-U” and “w/ reclassification”

In Table 1 abbreviations “Diag: diagnosis” and “FU: follow-up” are not used

In Figure 2 abbreviations “FU: follow-up” is not used

In Figure 3 add abbreviation: Bayesian model averaging models (BMA)

Figure 5 Add A and B in Figure

Author Response

We thank the reviewer for the feedback on our manuscript. The following minor points have been adapted in the manuscript.

Line 57 “during follow-up..” - Please check extra punctuation marks

We have corrected the punctuation.

In Table 1 words are far apart: “IBD-U” and “w/o reclassification”, “IBD-U” and “w/ reclassification”

We have adapted the result rows in Table 1 from jusitified to centred, thus it should be more readable.

In Table 1 abbreviations “Diag: diagnosis” and “FU: follow-up” are not used

The table legend has been adapted.

In Figure 2 abbreviations “FU: follow-up” is not used

The figure legend has been adapted.

In Figure 3 add abbreviation: Bayesian model averaging models (BMA).

The figure legend has been adapted.

Figure 5: Add A and B in Figure

The letters A and B were added to figure 5.

===================================

Round 2

Reviewer 1 Report

I appreciate the changes made by the authors that addressed most of my comments. However, I am still concerned about the 3 classes classification for CD, UC, and IBD-U. In Figure 4, with the color and shape switched version, we could clearly observe that the IBD-U is overlapped with the UC while both are separating from CD. I think this observation corroborates the re-classification results and the two-class argument. I would still suggest keeping this color scheme but using different shade areas, maybe with different line styles for the clusters in Figure 4. I understand that multi-class classification could be a harder task; however, the final precision and recall are just as important. It would be helpful if you could provide the full confusion matrices on the training and test sets and provide some discussions. Regarding the XGBoost results, do you also see the error rate go up for UC vs. IBD-U? If that's the case, would it be helpful to build a hierarchical classifier that first distinguishes the CD vs. non-CD, and then UC vs. IBD-U? Presumably, the features requested for these two tasks will vary.

I also think the supplementary table 3 showing the final penal can be a main table.

Author Response

Point-to-point Reply to Reviewer 1.

1) I appreciate the changes made by the authors that addressed most of my comments. However, I am still concerned about the 3 classes classification for CD, UC, and IBD-U. In Figure 4, with the color and shape switched version, we could clearly observe that the IBD-U is overlapped with the UC while both are separating from CD. I think this observation corroborates the re-classification results and the two-class argument. I would still suggest keeping this color scheme but using different shade areas, maybe with different line styles for the clusters in Figure 4.

We acknowledge the reviewer's concern about Figure 4 and agree with the notion that CD separates distinctly while UC and IBD-U overlap. As suggested, we replaced Figure 4 with a color and point shape switched version. (Clusters are now represented as point shapes, and clinical diagnoses are represented as colors.) To emphasize the separation of CD from the other two groups, we changed the method for the generation of the shaded areas to represent the Euclidean distance from group centers.

The Figure legend was adjusted accordingly:

“Figure 4. K-means clustering with k=3 pre-specified groups. The data points each represent one patient. Point shapes indicate the assigned cluster one to three; the colors indicate the clinical diagnosis of CD, UC, or IBD-U. Shaded areas indicate the Euclidean distance from the group center.”

2) I understand that multi-class classification could be a harder task; however, the final precision and recall are just as important. It would be helpful if you could provide the full confusion matrices on the training and test sets and provide some discussions. Regarding the XGBoost results, do you also see the error rate go up for UC vs. IBD-U? If that's the case, would it be helpful to build a hierarchical classifier that first distinguishes the CD vs. non-CD, and then UC vs. IBD-U? Presumably, the features requested for these two tasks will vary.

We thank the reviewer for this valuable input and provide for her/him with the full confusion matrices and performance metrics as well as the precision and recall.

Training and test metrics for the three-class logistic regression model:

Training and test metrics for the three-class XGBoost model:

Metrics for BMA, UC vs CD, not split:                      Precision and recall for log reg, XGBoost, and BMA:

Additionally, the reviewer suggests building a hierarchical model. Based on Figure 4, we assume that the separation of UC vs IBD-U is the most difficult task in this setting, thereby being the limiting factor for the performance of a hierarchical model. Therefore, we evaluated the binary classification between UC and IBD-U, implemented as a BMA model without train/test split. Please note that the p-value of the accuracy > non-information rate is 0.2, being not significant at alpha = 0.05. This indicates that the model is not able to reliably distinguish between UC and IBD-U. This is in accordance with Figure 4 showing a large overlap between IBD-U and UC. The full metrics are provided below.

3) I also think the supplementary table 3 showing the final penal can be a main table.

We thank the reviewer for this suggestion, we have added this table in the manuscript as table 1, and we have adapted the other table numberings accordingly